# Resveratrol Enhances Inhibition Effects of Cisplatin on Cell Migration and Invasion and Tumor Growth in Breast Cancer MDA-MB-231 Cell Models In Vivo and In Vitro

**DOI:** 10.3390/molecules26082204

**Published:** 2021-04-12

**Authors:** Meng-Die Yang, Yang Sun, Wen-Jun Zhou, Xiao-Zheng Xie, Qian-Mei Zhou, Yi-Yu Lu, Shi-Bing Su

**Affiliations:** 1Research Center for Traditional Chinese Medicine Complexity System, Institute of Interdisciplinary Integrative Medicine Research, Shanghai University of Traditional Chinese Medicine, Shanghai 201203, China; yangmengdie_yzu@foxmail.com (M.-D.Y.); bssunyang@126.com (Y.S.); be6835@126.com (W.-J.Z.); YearMonthDaySH@163.com (X.-Z.X.); tazhou@163.com (Q.-M.Z.); ava0048@163.com (Y.-Y.L.); 2WEIHAI WEGO BioTech CO., LTD., Weihai 264200, China

**Keywords:** resveratrol, cisplatin, breast cancer, MDA-MB-231 cells, epithelial–mesenchymal transition

## Abstract

Triple-negative breast cancer (TNBC) is a refractory type of breast cancer that does not yet have clinically effective drugs. The aim of this study is to investigate the synergistic effects and mechanisms of resveratrol combined with cisplatin on human breast cancer MDA-MB-231 (MDA231) cell viability, migration, and invasion in vivo and in vitro. In vitro, MTS assays showed that resveratrol combined with cisplatin inhibits cell viability as a concentration-dependent manner, and produced synergistic effects (CI < 1). Transwell assay showed that the combined treatment inhibits TGF-β1-induced cell migration and invasion. Immunofluorescence assays confirmed that resveratrol upregulated E-cadherin expression and downregulated vimentin expression. Western blot assay demonstrated that resveratrol combined with cisplatin significantly reduced the expression of fibronectin, vimentin, P-AKT, P-PI3K, P-JNK, P-ERK, Sma2, and Smad3 induced by TGF-β1 (*p* < 0.05), and increased the expression of E-cadherin (*p* < 0.05), respectively. In vivo, resveratrol enhanced tumor growth inhibition and reduced body weight loss and kidney function impairment by cisplatin in MDA231 xenografts, and significantly reduced the expressions of P-AKT, P-PI3K, Smad2, Smad3, P-JNK, P-ERK, and NF-κB in tumor tissues (*p* < 0.05). These results indicated that resveratrol combined with cisplatin inhibits the viability of breast cancer MDA231 cells synergistically, and inhibits MDA231 cells invasion and migration through Epithelial-mesenchymal transition (EMT) approach, and resveratrol enhanced anti-tumor effect and reduced side of cisplatin in MDA231 xenografts. The mechanism may be involved in the regulations of PI3K/AKT, JNK, ERK and NF-κB expressions.

## 1. Introduction

Breast cancer is the most common malignant tumor in women, and its incidence is on the rise [1,2]. It has become a major threat to women’s health. Triple-negative breast cancer (TNBC) is the most common and invasive breast cancer subtype in younger patients and is characterized by a lack of estrogen receptor, progesterone receptor, and human epidermal growth factor receptor 2 [3]. The lack of these receptors makes TNBC aggressive and does not respond to hormones and targeted therapies. In addition, TNBC is highly metastatic and can recur within three years [4].

In the past five years, the focus of TNBC treatment research has gradually focused on molecularly targeted drugs, including epidermal growth factor receptor antibodies, small molecule single targets, and multi-target tyrosine kinase inhibitors, anti-angiogenic, and DNA repair drugs [5]. Advances in research are expected to provide more treatment options for TNBC patients to improve cure rates and improve prognosis [6]. Although significant advances have been made in a variety of new drugs for HER2 or ER in recent years, the progression of TNBC is limited. Compared with other subtypes of breast cancer, TNBC patients have a survival rate of only 77% [7].

Resveratrol (trans-3,4,5-trihydroxystilbene) is a non-flavonoid polyphenol, derived from natural medicines such as *Polygonum cuspidatum, Rheum palmatum*, and fruits such as grapes, blueberries, mulberries and peanuts. It has been reported that resveratrol has the effect of anti-cancer, and can inhibit the occurrence and metastasis of breast cancer [8]. Previous studies showed that the anti-breast cancer effects of resveratrol includes inhibiting cell growth and proliferation by inducing autophagy [9] and apoptosis [10], reversing epithelial–mesenchymal transition (EMT) and decreasing metastasis [11,12], regulating the phase I and phase II detoxification system [13], affecting the epigenetic mechanism [14], increasing the sensitivity [11], and reducing cytotoxicity [15] of chemotherapy, suppressing multidrug resistance [16], and modulating immune response [17].

Cisplatin as a first-line drug for metastatic disease has a response rate of more than 40% in breast cancer metastasis [18]. It is a DNA-damaging drug, especially in the TNBC. Compared with other types of breast cancer, TNBC has a higher risk of distant recurrence and death in the first five years [19]. Due to the heterogeneity of TNBC, the lack of clear molecular targets [20] and the inherent genomic instability caused by TNBC lack of DNA repair may result in the production of platinum drugs (such as cisplatin or carboplatin) in the treatment of TNBC. In clinic, the addition of platinum drugs can significantly improve the pathological complete response rate in the neoadjuvant therapy of TNBC. However, cisplatin, one of the most active cytotoxic drugs at present, has a therapeutic effect on a variety of malignant tumors [21], and also produces serious side effects such as severe toxicity including nephrotoxicity [22], neurotoxicity [23], gastrointestinal toxicity [24], peripheral neuropathy [25], ototoxicity [26], and hematological toxicity [27]. Therefore, it is particularly necessary to find drugs that can reduce the side effects of cisplatin and enhance the therapeutic effects.

In this study, we investigated the synergistic effects of resveratrol combined with cisplatin on a TNBC models, MDA-MB-231 (MDA231) cell viability, migration and invasion in vivo and in vitro, and their effective mechanisms were also explored through EMT approach and the regulations of PI3K/AKT, JNK, ERK, and NF-κB signaling pathways.

## 2. Results

### 2.1. Resveratrol Combined with Cisplatin Inhibits Synergistically the Activity of MDA231 Cells

The effects of resveratrol combined with Cisplatin on the viability of MDA-MB-231 cells were detected by MTS assay. After treated with 2–64 μM Cisplatin (Figure 1A) and 12.5–250 μM resveratrol (Figure 1B) for 24, 48, and 72 h, cell viability decreased significantly (*p* < 0.05 or *p* < 0.01), and was positively correlated with time and concentration, showed a dose- and time-dependent manner. The IC50 were 185 μM in resveratrol combined with 14 μM in cisplatin. Furthermore, MDA231 cells were treated by 14 μM cisplatin was combined with resveratrol 50, 100, 150, 200, and 250 μM for 24 h, the survival rates of cells were 80.7%, 62.4%, 41.9%, 23.3%, and 10.4%, respectively, compared to those of resveratrol alone, the cell survival rate decreased by 16.7%, 25.1%, 19.0%, 21.8%, and 17.7% (Figure 1C). The CIs were 14 μM cisplatin combined with 200 μM resveratrol (Figure 1D). When MDA231 cells were treated by 185 μM resveratrol combined with cisplatin at 4, 8, 16, 32, and 64 μM for 24 h, the survival rates of cells were 80.2%, 61.6%, 24.2%, 17.3%, and 9.1%, respectively, compared to resveratrol alone (Figure 1E). The CIs were 175 μM resveratrol combined with 16 or 32 μM cisplatin. These results indicated that cisplatin was sensitized by resveratrol, high-dose resveratrol can enhance the efficacy of cisplatin in inhibiting tumor cell growth at low doses, and there are a synergistic effect of cisplatin and resveratrol on the viability of MDA231 cells.

### 2.2. Resveratrol Combined with Cisplatin Inhibits the Migration and Invasion of MDA231 Cells

The effects of resveratrol combined with cisplatin on the migration and invasion of MDA231 cells were detected by Transwell assay. As shown in Figure 2A,B, 12.5, 25 μM and 50 μM resveratrol combined with 4 μM cisplatin significantly inhibited the migration of MDA231 cells compared to control group or cisplatin group *(p* < 0.05 or *p* < 0.01), and the cell migration rate were 63.7%, 48.6%, and 28.3%, respectively. As shown in Figure 2C,D, 12.5, 25, 50 μM resveratrol combined with 4 μM cisplatin inhibited significantly the invasion of MDA231 cells, and the cell invasion rates were 66.5%, 61.3%, and 42.8%, respectively. The results showed that resveratrol combined with cisplatin can inhibit the migration and invasion of MDA231 cells.

### 2.3. Effect of Resveratrol Combined with Cisplatin on TGF-Β1-Induced Epithelial and Mesenchymal Molecular Markers in MDA231 Cells

In order to demonstrate the effect of resveratrol combined with cisplatin on the migration and invasion of MDA231 cells whether through EMT approach, TGF-β1-induced the changes of epithelial and mesenchymal molecular markers expressions by western blot and immunofluorescence assays. As shown in Figure 3A, the expressions of E-cadherin significantly decreased, while the expressions of vimentin and fibronectin increased by TGF-β1 induction (5 ng/mL) in MDA231 cells, compared to control group (*p* < 0.01). Moreover, TGF-β1 induced the expressions of E-cadherin significantly increased, while the expressions of vimentin and fibronectin decreased by resveratrol (12.5 μM, 25 μM, 50 μM), cisplatin (4 μM), and resveratrol combined with cisplatin treatments in MDA231 cells, compared to TGF-β1-treated group (*p* < 0.05).

Furthermore, the expression of EMT markers were verified by immunofluorescence. As shown in Figure 3C, 5 ng/mL TGF-β1 reduced the expression of E-cadherin and increased the expression of vimentin. When 50 μM resveratrol was combined with TGF-β1, the epithelial marker E-cadherin was increased and the mesenchymal marker vimentin was decreased. There was no significant change in the expression of E-cadherin and vimentin in the combination of 4 μM cisplatin and TGF-β1. The expression of E-cadherin was increased in the combination of resveratrol, cisplatin, and TGF-β1, and the expression of vimentin was decreased, which was consistent with the control group. These results indicated that the effects of resveratrol combined with cisplatin on the migration and invasion of MDA231 cells induced by TGF-β1 may be involved in the regulation of EMT.

Red represents E-cadherin and vimentin, and blue represents DAPI. Photographed under a laser confocal microscope (×625).

### 2.4. PI3K/AKT, Smad, NF-κB, JNK, ERK Signal Pathways May Involve in TGF-β1-Induced EMT by the Regulation of Resveratrol and Cisplatin in MDA231 Cells

In order to demonstrate the effect of resveratrol and cisplatin on the migration and invasion of MDA231 cells whether through EMT approach, 5 ng/mL TGF-β1-induced EMT was treated by 25 μM resveratrol, 4 μM cisplatin, and within or without 10 μM LY290042 (PI3K inhibitor), 10 μM SB431542 (Smad inhibitor), 10 μM PDTC (NF-κB inhibitor), 10 μM SP600125 (JNK Inhibitor), 10 μM PD98059 (ERK inhibitor) for 24 h and then the expression of epithelial and mesenchymal marker proteins were observed by western blot. As shown in Figure 4A, the expression of E-cadherin was decreased and the expression of vimentin and fibronectin were increased compared to control group (*p* < 0.01). When resveratrol and cisplatin were combined with LY290042 or SB431542 reversed the expression of these proteins induced by TGF-β1. The results showed that the effects of resveratrol and cisplatin on the EMT induced by TGF-β1 in MDA231 cells may be involved in the regulated of PI3K and Smad signaling pathways.

Furthermore, the resveratrol and cisplatin regulated the proteins expressions of EMT-related pathways in MDA231 cells were demonstrated by western blot. As shown in Figure 4B, the expression of P-AKT, P-PI3K, Smad2, Smad3, P-JNK, and P-ERK were increased in cells induced by TGF-β1. The combination of resveratrol 25 μM with TGF-β1 and cisplatin reversed the expression of these proteins and was superior to the treatment of cisplatin alone. The results indicate that the effects of resveratrol combined with cisplatin on the EMT induced by TGF-β1 in MDA231 cells may be involved in the regulation of PI3K/AKT and Smad, as well as related to the regulation of NF-κB, JNK, and ERK.

### 2.5. Resveratrol Enhances Anti-Tumor and Reduces Side Effects of Cisplatin in MDA231 Xenografts

In order to demonstrate the anti-tumor effect of resveratrol and cisplatin, MDA231 xenografts were prepared and the effects of resveratrol combined with cisplatin were assessed. As shown in Figure 5A, the effect of 50 mg/kg resveratrol combined with 5 mg/kg cisplatin was superior to cisplatin alone, which reduced significantly the tumor weight compared to model group (*p* < 0.01) and cisplatin-treated group (*p* < 0.05) in MDA231 xenografts from 3 to 8 weeks, while resveratrol alone was no effect. As shown in Figure 5B, the body weights of MDA231 xenografts were significantly increased by the combination treatments of resveratrol and cisplatin compared to cisplatin-treated group (*p* < 0.05), while the body weights were significantly decreased by cisplatin (*p* < 0.05). As shown in Figure 5C, serum BUN and Cr were decreased by the combination treatments of resveratrol and cisplatin compared to cisplatin-treated group (*p* < 0.05), while the serum BUN and Cr were significantly increased by cisplatin (*p* < 0.05). There were no significant changes of ALT and AST in each treatment group (*p* > 0.05). These results indicated that resveratrol can reduce body weight loss and kidney function impairment by cisplatin.

### 2.6. Resveratrol Combined with Cisplat Inhibits the Expression of P-AKT, P-PI3K, Smad2, Smad3, P-JNK, P-ERK, and NF-κB in Tumor Tissues of MDA231 Xenografts

In order to investigate the mechanisms of anti-tumor effect of resveratrol and cisplatin, the proteins expressions of tumor proliferation pathways in tumor tissues of MDA231 xenografts were demonstrated by western blot. As shown in Figure 6, the expressions of P-AKT, P-PI3K, Smad2, Smad3, P-JNK, P-ERK, and NF-κB were decreased by resveratrol and resveratrol combined with cisplatin compared to the model group, and the effects of the combination were better than in the resveratrol alone treatment. (*p* < 0.05), but there were no significant changes between cisplatin and model groups (*p* > 0.05). The results indicated that the regulations of P-AKT, P-PI3K, Smad2, Smad3, P-JNK, P-ERK, and NF-κB expressions may be involved in resveratrol enhances anti-tumor of cisplatin in MDA231 xenografts.

## 3. Materials and Methods

### 3.1. Reagents

MDA-MB-231 cell was purchased from the Shanghai Cell Bank of the Chinese Academy of Sciences (Shanghai, China). DMEM were purchased from Gibico (Brooklyn, NY, USA); MTS purchased from Promega (Madison, WI, USA); Smad2 and Smad3 were purchased from Santa Cruz; Fibronectin, P-PI3K, PI3K, P38, P-AKT, AKT, E-cadherin, Vimentin, Notch1, Wnt and NF-kB were purchased from Cell Signaling Technology (Danvers, MA, USA); LY290042, SB431542, SP600125 and PD98059 PDTC were purchased from Selleck Chemical Company (Houston, TX, USA); SB203580, Cyclopamin, ETC-159 and DAPT were purchased from MedChemExpresss (Monmouth Junction, NJ, USA); HRP-labeled goat anti-rabbit and anti-mouse IgG were purchased from Beijing Boda Tektronix Biogene Technology Co., Ltd. (Beijing, China); Resveratrol (98% content) was purchased from Shanghai Traditional Chinese Medicine Standardization Research Center (Shianghai, China). Cisplatin was purchased from Qilu Pharmaceutical (Hainan, China). IRDye^TM^ Fluorescent Antibody was purchased from Li-Cor Bioscience (Lincoin, NE, USA); Ultrasensitive Chemiluminescence Detection Kit was purchased from Shanghai Ya Enzyme Co., Ltd. (Shanghai, China); Immunofluorescence staining blocking solution (Containing DAPI) was purchased from Shanghai Biyuntian Company (Shanghai, China).

### 3.2. Cell Viability Assay

MDA231 cells of 5 × 10^7^/L. were inoculated 200 μL in each hole on 96-well culture plates. Resveratrol group with final concentration of 0, 12.5, 25, and 50 µM, cisplatin group of 10, 20, 40, 60, 80, 100 µM, 14 µM of cisplatin combined with resveratrol, 18 µM of resveratrol combined with cisplatin were set up according to the experimental requirements, each concentration has four compound holes. After 24, 48, and 72 h of incubation, 20 mL MTS was added to each well. The absorbance value (OD value) was measured at 490 nm. Calculate the combination indices (CIs) were analyzed using CompuSyn software (ComboSyn Inc., Paramus, NJ, USA). The CI value < 1 is a synergistic effect, the CI value = 1 is an additive effect, and the CI value > 1 is an antagonistic effect.

### 3.3. Invasion and Migration Assay

In the invasion experiment, 30 μg of Matrigel was applied to the upper layer of the chamber. Adjust the cell concentration to 1 × 10^5^/mL, add 200 μL of cell suspension with or without drug to the upper layer of the 24-well Transwell chamber. Aspirate the culture solution after 24 h, wipe off cells that have not invaded or migrated in the upper layer of the chamber by cotton swab. Rinse cells twice with PBS and fix with 4% paraformaldehyde for 15 min, then dye with 0.1% crystal violet for half an hour. Finally, the photo is counted and take semi-quantitative experiment.

### 3.4. Western Blot Assay

Whole cell lysates or tissue homogenate were electrophoresed, incubated and assayed as described in previous literature [12], The primary antibodies including E-cadherin, vimentin, fibronectin, AKT, P-AKT, PI3K, P-PI3K, Smad2, Smad3, ERK, P-ERK, P38, JNK, P-JNK, NF-κB, Wnt, Notch1 (all of 1:1000 attenuation), and secondary antibody GAPDH (1:5000 attenuation) were used. The assays were repeated at least three times.

### 3.5. Immunofluorescence Assay

MDA-MB-231 cells were adjusted to a cell density of 2.5 × 10^5^ cells/mL, and the cells were cultured on confocal dishes treated with TGF-β1 (5 ng/mL), TGF-β1 + resveratrol (50 μM), TGF-β1 + cisplatin (40 μM) or TGF-β1 + resveratrol + cisplatin for 24 h fixed with 4% paraformaldehyde for 30 min, and then stabilized in 0.5% Triton X-100 for 20 min. After three PBS washes and blocking with Quick Block^TM^ Blocking Buffer for Immunol Staining for 15 min, the cells were incubated with E-cadherin (1:200) and anti-Vimentin (1:500) antibodies overnight at 4 °C. After washing, the cells were blocked from light, incubated with an anti-rabbit antibody for 60 min and counterstained with DAPI. The cells were observed and photographed with a confocal fluorescence microscope (LSM880, Zeiss, Jena, Germany).

### 3.6. Preparation, Administration, and Treatment of MDA231 Xenografts

Seven-week-old female BALB/c mice (18–23 g) were fed at the Laboratory Animal Center at Shanghai University of Traditional Chinese Medicine. The preparation and administration of MDA231 Xenografts was as described in previous literature [12]. The mice were injected once every two days (days for Cisplatin) with 100 mg/kg resveratrol (resveratrol group, *n* = 10), 50 mg/kg cisplatin (Cisplatin group, *n* = 10) 100 mg/kg resveratrol +50 mg/kg cisplatin (Cisplatin group, *n* = 10) or PBS for MDA231 Xenografts (model group, *n* = 5) and PBS for BALB/c (normal group, *n* = 5) through the peritoneal cavity. Body of the mice were measured once per week. After 8 weeks of tumor cell inoculation, the mice were sacrificed, and the tumors were removed and weighed.

### 3.7. Liver and Kidney Function Tests

When the mice were sacrificed, 1 mL blood was collected from the eyes and then quickly centrifuged for 10 min at 3000 rpm to obtain the serum. The levels of serum ALT, AST, Cr, and BUN were detected according to the manufacturer’s colorimeter testing kits (Jiancheng Bioengineering Institute, Nanjing, China).

### 3.8. Statistical Analysis

The experimental data were expressed as mean ± standard deviation (x ± SD). The comparison of the two means was performed by *t*-test, and the comparison of multiple means was analyzed by variance. The difference was statistically significant at *p* < 0.05. Statistical analysis was performed by SPSS 19.0 software.

### 3.9. Ethics Approval and Consent to Participate

All animal procedures were conducted in accordance with the guidelines of the National Institutes of Health and were approved by the Ethical Committee of the Shanghai University of Traditional Chinese Medicine (approval ID PZSHUTCM18-101804).

## 4. Discussion

TNBC refers to breast cancer that does not express the estrogen receptor (ER), progesterone receptor (PR), and HER2/neu genes [28]. Although TNBC is sensitive to chemotherapy, such as the platinum can significantly improve the prognosis of TNBC, but the serious side effects of platinum drugs are also a non-negligible fact in clinical applications [29]. Therefore, it is particularly necessary to find new drugs or synergistic combination strategies that are more effective against TNBC and can alleviate the side effects of chemotherapy drugs.

Chinese herbal medicines (one of natural medicines) have long been used in cancer therapy as such synergistic combinations for enhancing efficacy, reducing side effects, immune modulation, as well as abrogating drug resistance of chemotherapy [30]. Therein, resveratrol-based combinatorial strategies for cancer management has also been increasingly studied, that showed resveratrol sensitizes tamoxifen in antiestrogen-resistant breast cancer cells [11] and TNF-β-induced survival of 5-FU-treated colorectal cancer cells [31], increases arsenic trioxide-induced apoptosis in chronic myeloid leukemia cells [32] and lung cancer cells [33], decreases cytotoxicity of doxorubicin in breast cancer cells [15], and combined with piceatannol upregulates PD-L1 expression in breast and colorectal cancer cells [34]. However, the effects of resveratrol combined with cisplatin on breast cancer is still unclear.

In this study, we demonstrated the effects of resveratrol combined with cisplatin using a TNBC models, MDA231 cells in vivo and in vitro. Our results showed that, in vitro, resveratrol combined with cisplatin inhibits synergistically cell viability, and inhibits TGFβ1-induced cell migration and invasion (Figure 1 and Figure 2); in vivo, resveratrol enhanced tumor growth inhibition and reduced body weight loss and kidney function impairment by cisplatin in MDA231 xenografts (Figure 5). These results indicated that resveratrol may sensitize the inhibitive effects of cisplatin on cell viability, migration and invasion, and tumor growth in MDA231 models in vivo and in vitro.

EMT is a process in which epithelial cells lose cell polarity and intercellular adhesion, gain migration, and invasiveness and become interstitial cells. Epithelial cells express high levels of E-cadherin, while mesenchymal cells express N-cadherin, fibronectin, and vimentin. Thus, EMT causes morphological and phenotypic changes in cells [35]. EMT plays an important role in the process of TNBC. It has been reported that resveratrol can reverse EMT [12], and sensitizes tamoxifen in antiestrogen-resistant with EMT in breast cancer cells [11]. In this study, we found that resveratrol sensitizes the effects of cisplatin inhibits TGFβ1-induced MDA231 cell migration and invasion by reducing the expression of vimentin, fibronectin, and increased the expression of E-cadherin, which is reversing EMT, while there is no obvious effect in those of cisplatin alone (Figure 3). It indicated that resveratrol gives cisplatin an efficacy of reversing EMT.

There are multiple dysregulated signaling pathways such as Wnt/β-catenin, Notch, NF-κB, PI3K/Akt, Smad, MAPK (including p38, JNK and ERK) and Hedgehog in TNBC, and these signaling pathways involved in the regulation of cell growth, proliferation, migration, EMT and metastasis, and activation of apoptosis, and affected by natural compounds such as resveratrol or its combination with classical chemotherapeutic agents in TNBC [36]. In order to clarify which signaling pathway involved in the regulation of EMT treated by resveratrol and cisplatin combination in MDA231 cells, we screened the TGF-β1-induced EMT-related signaling pathways, using inhibitors including LY290042 for PI3K, SB431542 for Smad, PDTC for NF-κB, SP600125 for JNK, and PD98059 for ERK. Our results showed that, PI3K and Smad signaling pathways may be involved in the regulated of EMT induced by TGF-β1 in MDA231 cells (Figure 4A). Moreover, further experiments showed the combination of resveratrol and cisplatin downregulated the expression of P-JNK, and P-ERK, which increased in MDA231 cells induced by TGF-β1 (Figure 4B), indicated the effects on EMT were involved in the regulation of PI3K/AKT and Smad signal pathways, as well as regulated NF-κB, JNK, and ERK expressions.

Moreover, previous studies have reported that resveratrol inhibits tumor growth by inducing apoptosis in MDA231 xenograft and HER-2/neu transgenic mice models [37,38]. The inhibitive tumor effects of resveratrol combined with quercetin and catechin by the regulation of cell cycle progression in MDA231 xenograft has also been reported [39]. However, the tumor-inhibited mechanisms of resveratrol combined with cisplatin on breast cancer is still unclear. In this study, we found that resveratrol enhanced the inhibitive effects of cisplatin on P-AKT, P-PI3K, Smad2, Smad3, P-JNK, P-ERK, and NF-κB expressions in MDA231 xenografts (Figure 6) indicated that the regulations of P-AKT, P-PI3K, Smad2, Smad3, P-JNK, P-ERK, and NF-κB expressions may be involved in resveratrol enhances anti-tumor of cisplatin in MDA231 xenografts. In addition, because resveratrol can inhibit tumor metastasis [12] and multiple signal pathway involved in metastasis of breast cancer [8,36], further study will investigate the effects and mechanisms of Resveratrol combined with cisplatin on the tumor metastasis.

## 5. Conclusions

In summary, resveratrol combined with cisplatin produced a synergistic effect on the inhibition of breast cancer cell viability, inhibits breast cancer MDA231 cell migration and invasion through EMT regulated by PI3K/AKT, Smad, NF-κB, JNK, and ERK. Moreover, resveratrol enhanced anti-tumor effect and reduced side of cisplatin in MDA231 xenografts and the effective mechanism may be involved in the regulations of PI3K/AKT, JNK, ERK, and NF-κB expressions.

## Figures and Tables

**Figure 1 molecules-26-02204-f001:**
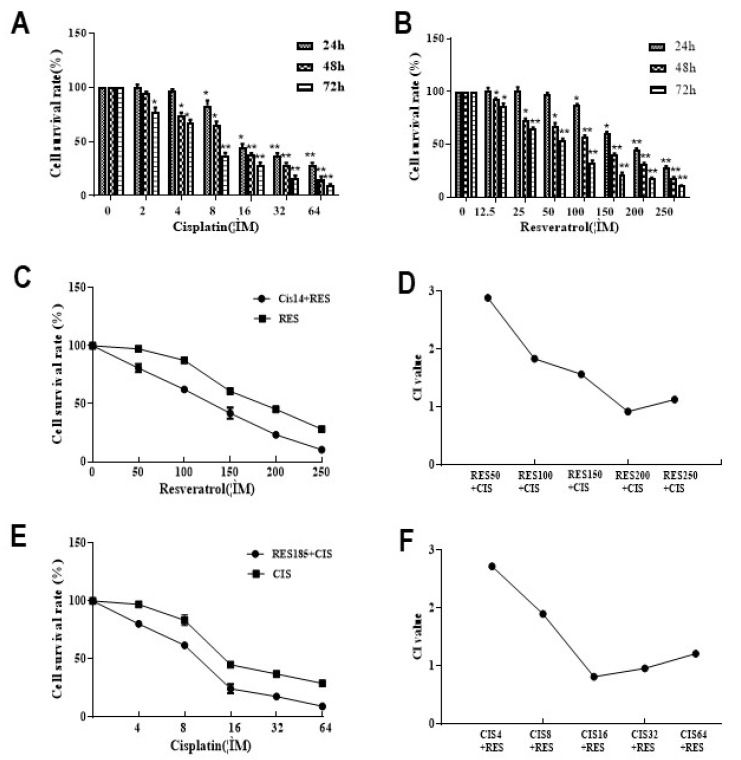
Effect of cisplatin and resveratrol on the viability of MDA231 cells. (**A**) Cell survival rate treated by cisplatin; (**B**) cell survival rate treated by resveratrol for 24, 48, and 72 h; (**C**) cell survival rate treated by 14 μM cisplatin combined with different concentrations of resveratrol for 24 h; (**D**) CI value of cisplatin 14 μM combined with resveratrol treatment; (**E**) cell survival rate treated by 185 μM resveratrol combined with different concentrations of cisplatin for 24 h; (**F**) CI value of 185 μM resveratrol combined with cisplatin treatment. The experiment was repeated three times, RES, resveratrol. CIS, cisplatin. * *p* < 0.05, ** *p* < 0.01, vs. the control group.

**Figure 2 molecules-26-02204-f002:**
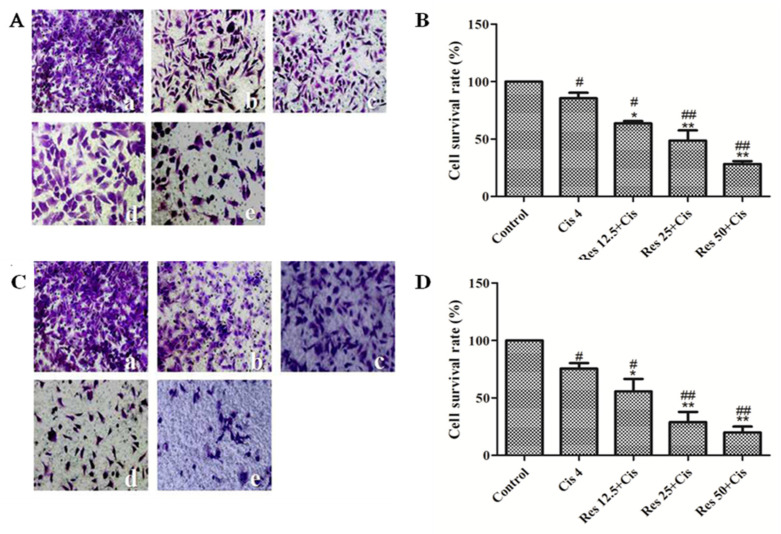
Effects of resveratrol combined with cisplatin on the migration and invasion of MDA231 cells. Transwell chamber migration and invasion experiments (×200). (**A**) Transwell chambers images of cell migration; (**B**) cell migration rate; (**C**) transwell chambers images of cell invasion; (**D**) cell invasion rate. (a) Control group; (b) 4 μM cisplatin group; (c) 12.5 μM resveratrol + 4 μM cisplatin group; (d) 25 μM resveratrol + 4 μM cisplatin group; (e) 50 μM resveratrol + 4 μM cisplatin group. # vs. control group; * vs. cisplatin group. */# *p* < 0.05, **/## *p* < 0.01, vs. the control group. Error bars represent three independent experiments, each performed three times.

**Figure 3 molecules-26-02204-f003:**
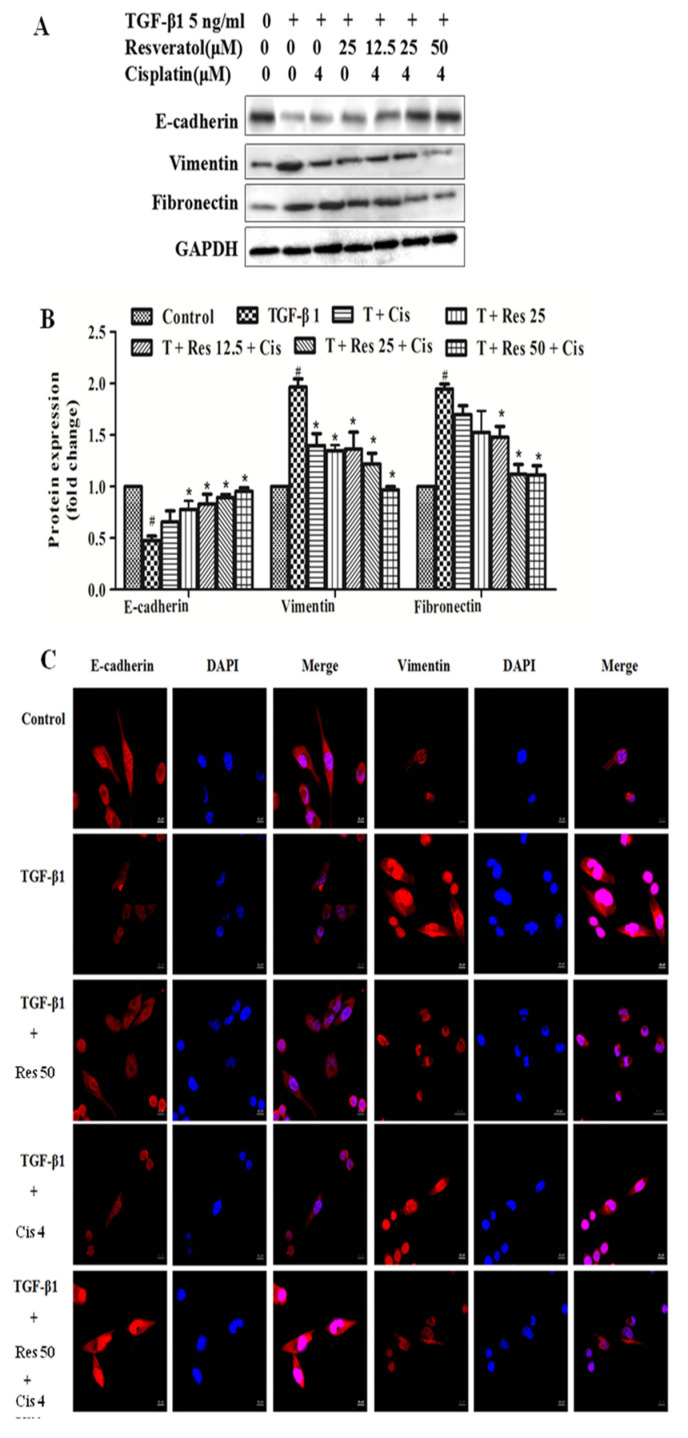
Regulatory effect of resveratrol combined with cisplatin on TGF-β1-induced epithelial and mesenchymal molecular markers in MDA231 cells. (**A**) TGF-β1-induced expressions of E-cadherin, vimentin and fibronectin treated by 5 ng/mL TGF-β1, 12.5 μM, 25 μM, 50 μM resveratrol, 4 μM cisplatin or resveratrol + cisplatin + TGF-β1 for 24 h. (**B**) Protein expression fold change. The experiment was repeated three times, # *p* < 0.01, vs. control group; * *p* < 0.05, vs. TGF-β1-treated group. (**C**) TGF-β1-induced expressions of E-cadherin and vimentin with immunofluorescence assay, treated by 50 μM resveratrol and 4 μM cisplatin for 24 h.

**Figure 4 molecules-26-02204-f004:**
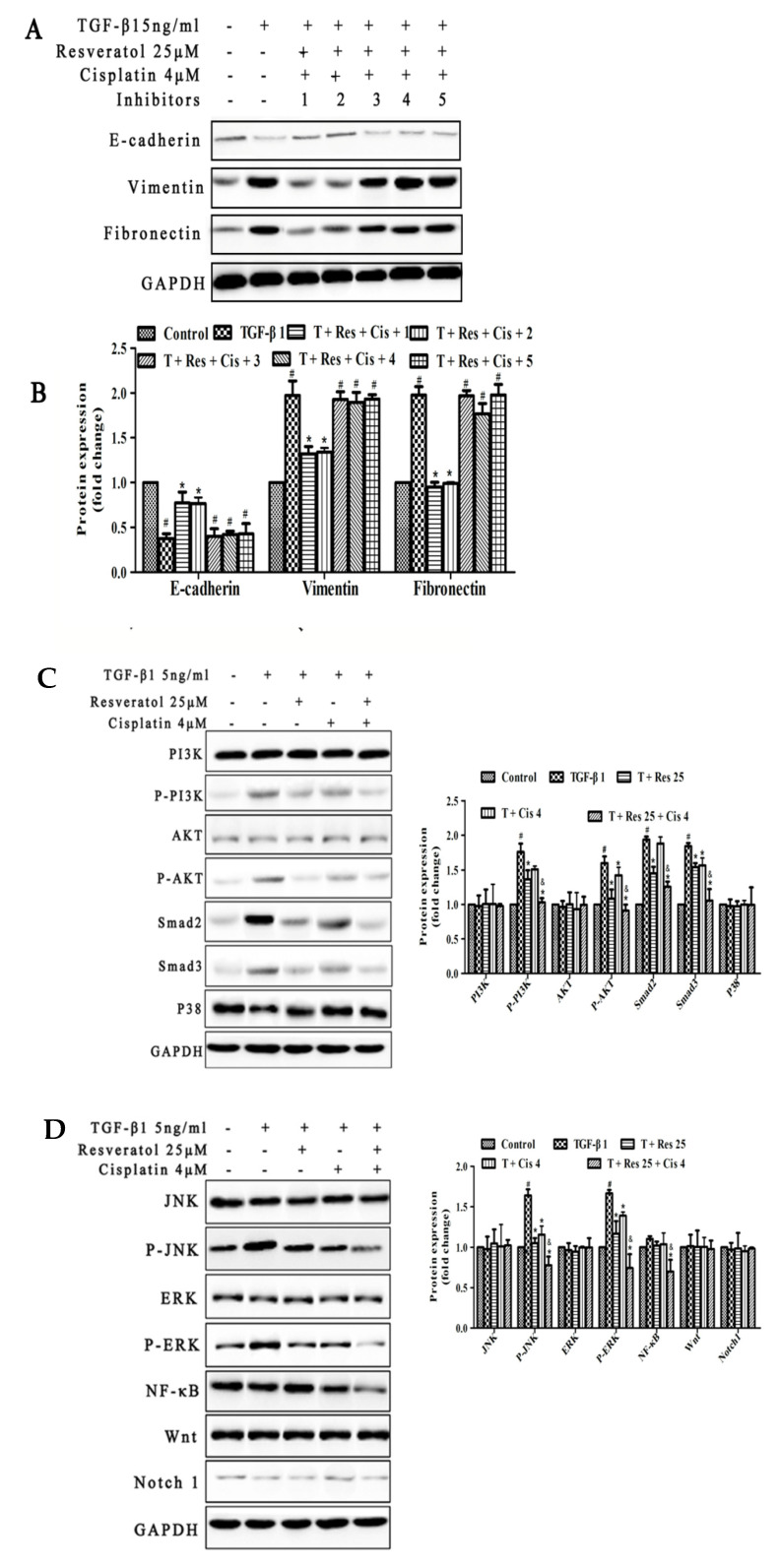
Effects of resveratrol and cisplatin on EMT-related pathway proteins induced by TGF-β1 in MDA231 cells. (**A**) Expressions of epithelial and mesenchymal markers by the treatment of signaling pathway inhibitors; (**C**) Expressions of EMT-related pathway proteins (**B**,**D**) protein expression fold change. T, TGF-β1 5 ng/mL; 1, LY290042; 2, SB431542; 3, SP600125; 4, PD98059; 5, PDTC. The experiment was repeated three times, # *p* < 0.01, vs. control group; * *p* < 0.05, vs. TGF-β1-treated group; & *p* < 0.01, compared to TGF-β1 + cisplatin-treated group.

**Figure 5 molecules-26-02204-f005:**
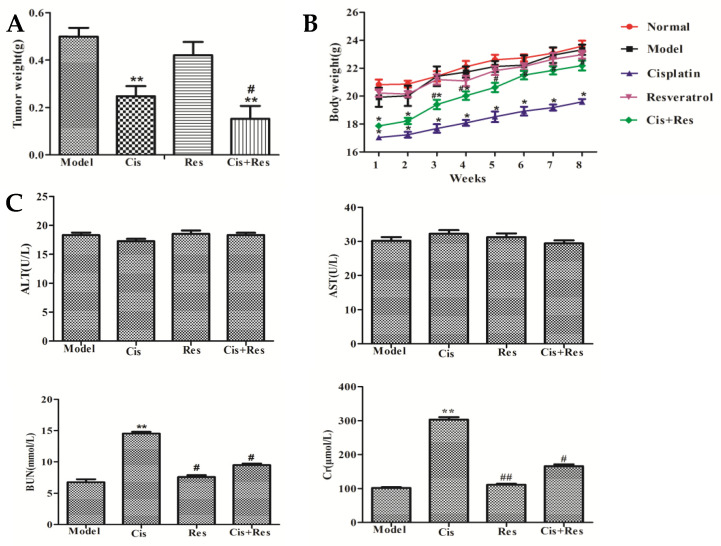
Anti-tumor effect of resveratrol and cisplatin without side effects in MDA231 xenografts. (**A**) Effects on tumor weight; (**B**) effects on body weight; (**C**) effects on serum ALT, AST, BUN, and Cr. ** *p* < 0.01, ** p* < 0.05 vs. model group; ## *p* < 0.01, # *p* < 0.05, vs. cisplatin-treated group.

**Figure 6 molecules-26-02204-f006:**
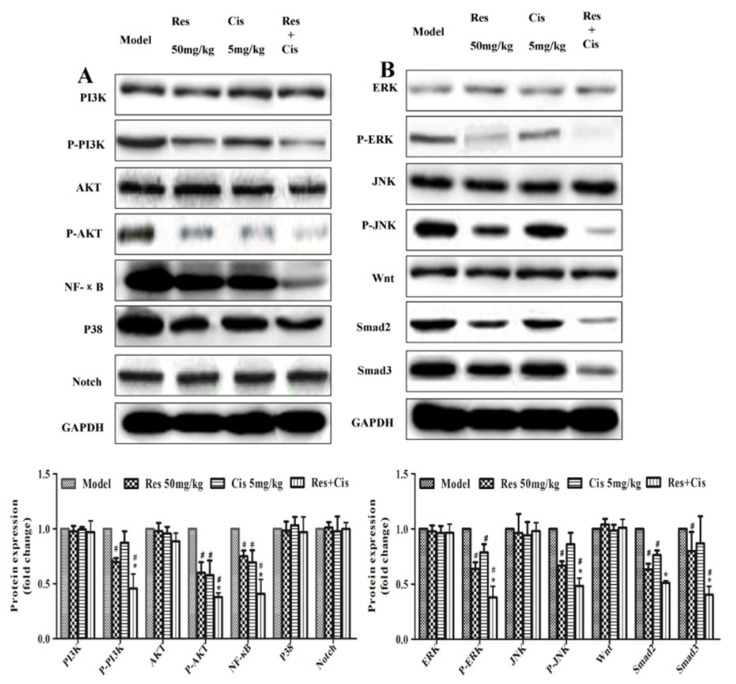
Effect of resveratrol and cisplatin on the expression of P-AKT, P-PI3K, Smad2, Smad3, P-JNK, P-ERK, and NF-κB in tumor tissues of MDA231 xenografts. (**A**) Fragments of each protein expressions by western blot analysis; (**B**) Fold changes of protein expressions. The experiment was repeated three times. ** p* < 0.05 vs. model group; # *p* < 0.05, vs. cisplatin-treated group.

## Data Availability

The data presented in this study are available with the authors.

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
