# Peer review of "Resveratrol Enhances Inhibition Effects of Cisplatin on Cell Migration and Invasion and Tumor Growth in Breast Cancer MDA-MB-231 Cell Models In Vivo and In Vitro"

_molecules, 2021, doi:10.3390/molecules26082204_

Round 1

Reviewer 1 Report

The manuscript in reference describes the study focused on the synergistic effect of Resveratrol combined with Cisplatin on human breast cancer MDA-MB-231 (MDA231) in vitro and in vivo. The manuscript is interesting and has important results. however, some points must be addressed prior further consideration.

  1. A detailed scrutiny about the language style and grammar should be performed throughout the manuscript to improve its quality.
  2. Line 51: Botanical names, Polygonum cuspidatum, Rheum palmatum, must be written in italics.
  3. According to Molecules' Authors Guide, Results and Discussion section should be placed at ending of manuscript.
  4. Line 96: the volume unit must be "μL" instead of "μl".
  5. Figure 4: some letters are superimposed on chart's information.
  6. Line 335: A explanation and clarification of this idea must be provided regarding how resveratrol could give cisplatin efficacy of reversing EMT.
  7. Lines 342-351: The information included in these lines should be improved, since they are only limited to describe results but an adequate discussion is not provided. For instance, a comparison with other describing such expression with other phenolics is missing.
  8. Figure 6a: The resolution of the western blot recording of P-AKT must be improved.
  9. The synergistic effects regarding concentration combinations of Cis and RES, presented in Figure 1, are not adequate discussed and their scope explored. 
  10. A disconnection is found between results in section 2.1 with others, since the concentrations of Cis and RES are not justified in each section, so the results are not totally consistent. I recommend to justify the concentration used in each section.

Reviewer 2 Report

In this manuscript, Yang et al. studied the effect of the combination of resveratrol and cisplatin on the MDA-MB-231 triple negative breast cancer cell line. They have addressed cell viability, migration and invasion in vitro, tested the levels of several proteins involved in this processes related to cancer progression and also in the epithelial-to-mesenchymal transition (EMT), after stimulating the cells with TGF-b1, and finally, performed in vivo experiments in MDA-231 xenografts to demonstrate that resveratrol enhances the antitumor efficacy of cisplatin both in vivo and in vitro.

Although there are many relevant results in the manuscript that make it interesting and suitable for publication in Molecules, there are several points to be solved; Therefore, I consider that a deep major revision is necessary before considering the manuscript for publication:

Firstly, the authors shall improve the presentation of the manuscript and do some editing to the language. Additionally, the authors shall organize better the typography of the manuscript. Some examples are:

Abstract line 12: “is a refractory type in breast cancer, and lack of effective drugs clinically”: Is a refractory type of breast cancer that does not yet have clinically effective drugs

Line 19: “E-cadherin expression and up down-regulated Vimentin…”

Figure 2, lines 194-196: revestrol instead of resveratrol.

Figure 4A: under the GAPDH panel, a half panel is seen, presumably corresponding to another protein that the authors did not consider relevant for the manuscript.

In the same figure 4, C and D are overlapped with the figure.

Figure legends are not justified

Double spaces: lines 13, 170, 201

Sentence in lines 200-203 has no sense, same line 206… and so on many times thoughout the manuscript…

Concerning the results:

Fig 1B: the results of resveratrol 0-100 mM of the figure are identical to those previously published (Molecules, 2019, 24(6) 1131. Not similar, but identical.

Fig 2B and 2D, Cell survival rate? Perhaps the authors mean Percent of cells proliferation instead?

Lines 209-21: if there is no significant difference, authors cannot state that effects of combination were better than those of cisplatin and resveratrol alone.

Figure 4A: I really do not understand this figure. TGF-b1 induces both vimentin and fibronectin and decreases E-cadherin as expected (column 2). Columns 3, 4 and 5 should be TGF plus resveratrol (+), TGF plus Cisplatin (+) and TGF plus both respectively. Additionally, TGF in combination with each one of the inhibitors in absence of cisplatin and resveratrol should be included. In my interpration, if resveratrol plus cisplatin effectively inhibit the expression of vimentin, the inhibitory effect is abolished by inhibitors 3, 4 and 5. Why did the authors use 25 mM resveratrol in figure 4, when figure 3 proves that 50 mM is much more effective?

In the previous study published in Molecules, the authors have described effects of resveratrol on Snail1 and Slug protein levels. Since the present study claims that resveratrol enhances the effects of cisplatin on cell migration and invasion, information about these proteins involved in the EMT transition is relevant.

Reviewer 3 Report

I have a few suggestion and questions for authors to pursue:

In figure 1 (C-F), add the error bars in the line graph to show the significance of the data.

What was the negative control (normal cell line) used in the experiments?

Round 2

Reviewer 3 Report

The authors have successfully addressed my concerns...